# Dark matter directionality approach using ZnWO$_4$ crystal scintillators

Vincenzo Caracciolo[1,2,*], Volodymyr Yakovych Degoda[3], Pierluigi Belli[1,2],
Rita Bernabei[1,2], Y. A. Borovlev[4,5], Fabio Cappella[6,7], Riccardo Cerulli[1,2],
Fedor A. Danevich[2,8], Antonella Incicchitti[6,7], Alice Leoncini[1,2], Vittorio Merlo[1,2],
N. Cherubini[9], Dmytro V. Kasperovych[8], Y. P. Kogut[3], G. P. Podust[3],
Oksana G. Polischuk[6,8], A. G. Postupaeva[5], V. N. Shlegel[4] and Volodymyr I. Tretyak[8,10]

**1** Dipartimento di Fisica, Università di Roma "Tor Vergata", I-00133 Rome, Italy
**2** INFN, sezione di Roma "Tor Vergata", I-00133 Rome, Italy
**3** Taras Shevchenko National University of Kyiv, 01601 Kyiv, Ukraine
**4** Nikolaev Institute of Inorganic Chemistry, 630090 Novosibirsk, Russia
**5** CML Ltd, 630090 Novosibirsk, Russia
**6** INFN, sezione di Roma, I-00185 Rome, Italy
**7** Dipartimento di Fisica, Università di Roma "La Sapienza", I-00185 Rome, Italy
**8** Institute for Nuclear Research of NASU, 03028 Kyiv, Ukraine
**9** ENEA, Italian National Agency for New Technologies, Energy and Sustainable Economic Development, C.R. Casaccia, 00123, Rome, Italy
**10** INFN, Laboratori Nazionali del Gran Sasso, 67100 L'Aquila, Italy

⋆ vincenzo.caracciolo@roma2.infn.it

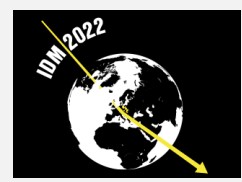 *14th International Conference on Identification of Dark Matter
Vienna, Austria, 18-22 July 2022*

## Abstract

The development of low-background anisotropic detectors can offer a unique way to study those Dark Matter (DM) candidate particles able to induce nuclear recoils through the directionality technique. Among the anisotropic scintillators, the ZnWO$_4$ has unique features and is an excellent candidate for the purposes. Both the light output and the scintillation pulse shape depend on the impinging direction of heavy particles with respect to the crystallographic axes and can supply two independent modes to study the directionality and discriminate $\gamma/\beta$ radiation. Measurements to study the anisotropic and scintillation performances of ZnWO$_4$ are reported.

# 1 Introduction

Astrophysical studies have suggested the presence of Dark Matter (DM) on all the astrophysical scales. Many statements have indicated that a significant fraction of it should be in the form of relic particles.

Nowadays, several approaches are used to study DM particles in the galactic halo; here, we will concentrate on a precise strategy: the directionality approach.

In direct detection experiments, a model-independent signature is a robust mechanism to furnish a DM signal marker with respect to the background. Actually, as initially suggested in Refs. [1, 2] and examined in Refs. [3–5], the events induced by a physical interaction in a target detector are linked with the halo model, the cross-section of the process and the relative velocity between the incident DM particle (DMp) and the target. Such an interaction rate is expected to have a characteristic time behaviour, namely the DM annual modulation, successfully exploited by DAMA collaboration [6–12, 14, 15].

Aside from this main signature, other potential ones are predictable: a diurnal modulation, due to the Earth revolution around its axis [16]; a daily variation of the interaction rate due to the different Earth depth crossed by the DMp [17] and the directionality signature, due to the correlation of DM impinging direction with Earth's galactic motion [18]. In fact, the rotation dynamics of the Milky Way galactic disc through the halo of DM induces the Earth to encounter a wind of DM particles that apparently flows along a path contrary to the solar motion with respect to the DM halo.

Thus, observing an anisotropy in the distribution of nuclear recoil directions will give additional proof and details for some DMp candidates and related astrophysics scenarios. Therefore, a direction-sensitive detector is needed.

In the case of the tracking detector, several R&D's for experiments based on Low-Pressure TPC, requiring a suitable mass and angular resolution, are in progress. Another approach uses a nuclear emulsion-based detector acting both as a target and as a solid tracking device [19]. We also mention two other ideas. A proposed method identifies the direction of an incident DMp using the spectroscopy of quantum defects in macroscopic solid-state crystals as diamonds [20]. Instead, in Ref. [21], a type of DM detector made of DNA (or RNA) is proposed.

Another main group of detectors is based on the anisotropic response, showing a signal that varies with the direction of impinging particles. In particular, in literature, there are attempts to use (i) carbon nanotubes, (ii) columnar recombination in a TPC dual-phase, (iii) anisotropic crystal scintillators. At the moment, there are weak indications about the possibility to use the first two ideas (see [22], [23], and Fig. 21 of Ref. [24]). However, the anisotropic scintillation detectors are a very promising technique, as described in the following. In addition, such a technique overcome some limitations of the other mentioned experimental attempts, as, for example, the mass (and thus the volume) needed in the case of Low-Pressure TPC and nuclear emulsions, the required angular resolution at very low energy of tracking detectors, the operation stability for a suitable long time of data taking, etc.

# 2 Study of the ZnWO$_4$ anisotropic response

In so-called anisotropic scintillators, the detector response to highly ionising particles (in terms of quenching factor, Q.F., and pulse shape) depends on their impinging direction with respect to the crystallographic axes. Accordingly, in the case of nuclear recoils, the energy spectrum shape at low energy is expected to vary during the sidereal day due to the changing orientation of the crystallographic axes relative to the DMp direction. These characteristics induce a peculiar

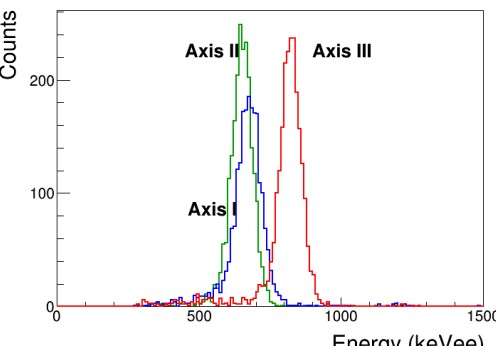
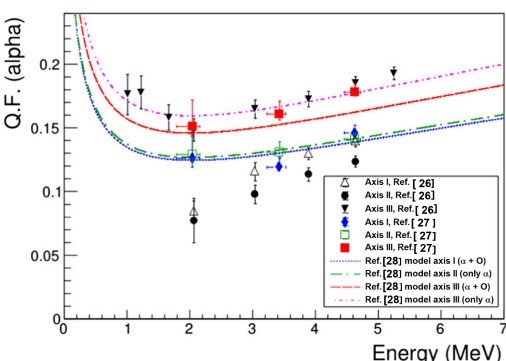

Figure 1: Left: energy spectra of the $\alpha$ particles with 4.63 MeV with the hitting direction parallel to the crystallographic axes of $ZnWO_4$ crystal scintillator. Right: dependence of the Q.F. versus the energy of the $\alpha$ particles measured with a $ZnWO_4$ scintillator in Ref. [26] (black points) compared with those reported in Ref. [27] (coloured points). The plot shows an evident anisotropic Q.F. for $ZnWO_4$ detectors. Following the prescription of Ref. [28], the analytic behaviour for each crystallographic axis, considering a global fit on all the data (from $\alpha$'s and recoils; see also later) of Ref. [27], is also reported. These figures are being reused from Ref. [27] with permission (license number 5396390679264).

variation of the counting rate measured in a given low energy window along the sidereal day and offer the possibility to emphasise the DM events to the electromagnetic background (for details, see, e.g., in Refs. [3–5, 18, 25]).

The crystal scintillator detectors with anisotropic response were suggested in Ref. [3] and reexamined in Ref. [4] in order to study the directionality signature. Initially, the anthracene scintillator was studied; however, several practical difficulties, in the workability of such scintillators, were pointed out. To overcome these issues $ZnWO_4$ crystal scintillator were suggested in [18].

The first measurements of the Q.F.'s of $ZnWO_4$ and the related anisotropic effect were studied in Ref. [26] using $\alpha$ particles. Later, in Ref. [27], a small $ZnWO_4$ crystal and an $^{241}$Am source with various sets of thin mylar films to decrease the energy of the $\alpha$ particles, have been used to perform new measurements. The energy scale of the crystal, for each measurement, was calibrated using $^{137}$Cs and $^{22}$Na $\gamma$ sources.

In Fig. 1-left is reported the energy distributions of $\alpha$ particles hitting the crystal along the three crystallographic axes (010), (001) and (100): hereafter III, II and I respectively in Fig. 1. In Fig. 1-right, the dependence of the Q.F. as a function of the energy for the three different directions of the $\alpha$ beam related to the crystallographic axes is illustrated. In particular, the Q.F. for $\alpha$ particles measured along the crystallographic axis III is $\sim 1.2$ times larger than that measured along the crystallographic axes I and II. Instead, the Q.F.'s estimates along the crystallographic axes I and II are reasonably comparable. The Q.F.'s reported in Ref. [27] are in suitable congruity with those of Ref. [26], as pointed out in Fig. 1-right. In the same figure, the shape of the Q.F. versus the energy for each crystallographic axis, as predicted in the model of Ref. [28], is also reported. Consequently, the data support the anisotropic characteristics of $ZnWO_4$ crystal scintillator for the $\alpha$ rays.

Using the same crystal and adopting a monochromatic neutron generator, the energies of the O nucleus recoiled have been investigated. In particular, a scheme of the set-up is illustrated in Fig. 4 of Ref [27], and the detailed data analysis are described there. Fig. 2 shows the obtained values of the Q.F.'s and the models for the studied crystallographic axes

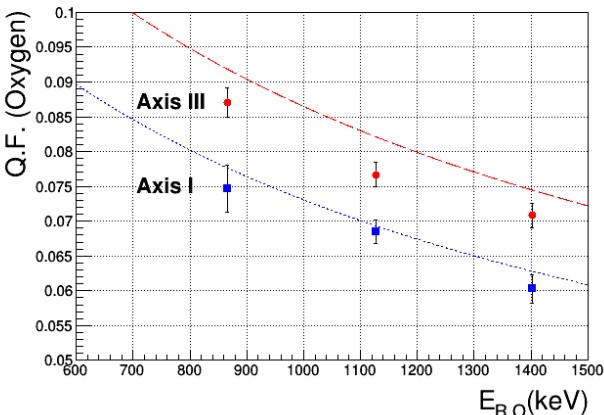

Figure 2: Q.F.'s for O nuclear recoils in ZnWO$_4$ for the crystallographic axes I and III as a function of the expected recoil energies $E_{R,O}$. The model of the Q.F. expected for the considered crystallographic axes is also reported; it has been evaluated using the global fits of the $\alpha$ and O recoil data, considering the indications of Ref. [28] (for more details, see Ref. [27]). This figure is being reused from Ref. [27] with permission (license number 5396390679264).

deduced by Ref. [28] (see Ref. [27] for details). In Ref [27], we point out that the anisotropy is significantly clear also for oxygen nuclear recoils in the energy region down to $\sim 100$ keV at 5.4 $\sigma$ of the C.L. (see also Tab. 1 of Ref. [27]).

## 3 Optical and scintillation properties of advanced ZnWO$_4$ crystal scintillators

Thanks to the low-thermal gradient Czochralski technique and an extended R&D, increased optical and scintillation features of ZnWO$_4$ crystals were obtained. A central part of the R&D was performed considering different compound stoichiometry of the initial WO$_3$, coming from several producers and additionally purified. Moreover, several investigations were performed considering a single or a double crystallization of the crystal and implemented or not implemented the annealing of the grown boules. The thermally stimulated luminescence, the emission spectra, phosphorescence, temperature and the dose dependencies of the luminescence intensity under X-ray irradiation of the crystal samples were studied from 85 K to room temperature. In particular, the thermally stimulated luminescence was studied up to 350 K. The scintillation characteristics of the ZnWO$_4$ crystals were studied using the following *gamma* quanta sources: $^{241}$Am, $^{137}$Cs, $^{207}$Bi, $^{60}$Co and $^{232}$Th. The ZnWO$_4$ crystals, developed by a single crystallization using a ZnWO$_4$ compound with the stoichiometric composition, with a deep purification of the WO$_3$ and implemented an annealed procedure in the air atmosphere of the crystal, have been shown the best crystals' optical transmission spectra in the range of 300-700 nm (see Ref. [29] for more details).

Fig. 3 shows the measured energy spectrum using $\gamma$ quanta from $^{241}$Am and $^{137}$Cs sources with the ZnWO$_4$ crystal sample that exhibited the highest scintillation properties (see Ref. [29]). The full-width half-maximum (R in the Fig. 3) is the best one documented in the literature for a ZnWO$_4$ crystal scintillator. The scintillation light output and the luminescence intensity of the ZnWO$_4$ samples (which can be justified by a dose dependence of XRL intensity and the phosphorescence behaviour, both marginal in scintillation measurements) don't show

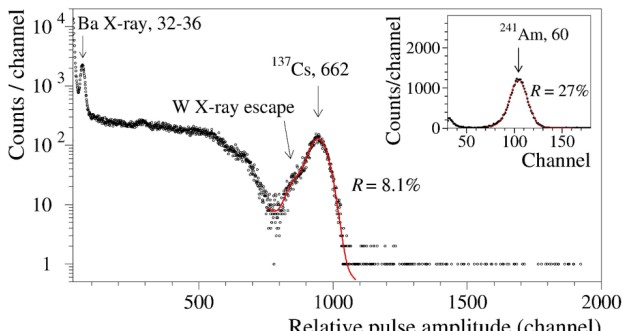

Figure 3: The energy spectra of $\gamma$-ray quanta of $^{137}$Cs measured by scintillation detector with the ZnWO$_4$ crystal sample No. 84 (sample sizes: 10 mm × 10 mm × 2 mm) of Ref. [29]. Energy spectrum of $\gamma$-ray quanta of $^{241}$Am is shown in inset. The energy of quanta lines is in keV. This figure is being reused from Ref. [29] with permission (license number 5396390086286).

a clear correlation. This absence of correspondence between the scintillation light output and the luminescence intensity indicates that the achieved scintillators' quality (especially for the double recrystallized samples) is not perfect; thus, there is some possibility to improve the ZnWO$_4$ production technology. A dedicated R&D is in progress to develop larger volume ZnWO$_4$ scintillators for rare event experiments.

## 4 How to profit of anisotropic scintillators and conclusions

As a consequence of the anisotropy light response for highly ionising particles, recoil nuclei induced by the considered DM candidates could be discriminated from the background thanks to the expected variation of their low energy distribution along the day. Thus, the expected signal counting rate in the energy window of interest is a function of the time. Assuming a multi detectors set-up and using a matrix of, e.g., 5 × 5 ZnWO$_4$ crystals for a total mass of 200 kg, 5 years of data taking, a software energy threshold of 2 keV and a simplified framework described in Ref. [18],[1] the considered experiment can reach a sensitivity, for the cross section, at level of $10^{-5} - 10^{-7}$ pb, depending on the DMp mass and the background level from $10^{-4}$ to 0.1 cpd/kg/keV. For a complete and detailed discussion see Ref. [18]. In Fig. 4, an expected counting rate as a function of sidereal time and days of the year in case of a DMp-nucleus elastic scattering with a multi detectors of ZnWO$_4$ anisotropic scintillators is presented. The model assumes a DMp mass of 10 GeV, with a cross section on nucleon of $5 \times 10^{-5}$ pb (dominated by a spin-independent coupling constant and a scaling law, of the DM-nucleus elastic cross section, very simple, see Ref. [18] for all the model details).

In conclusion, the directionality DM experiments could obtain, with a different new approach, further evidence for the presence of DM candidates inducing nuclear recoils in the galactic halo and/or provide complementary information on the nature and interaction type of DMp candidates. The anisotropic ZnWO$_4$ detectors are promising to investigate the directionality for DM candidates inducing nuclear recoils. An extensive R&D has been performed to obtain very high quality of ZnWO$_4$ crystal scintillators. First evidences of anisotropy in the response of ZnWO$_4$ crystal scintillator to nuclear recoils have been reported in Ref. [27] in the energy region down to some hundreds keV at 5.4 $\sigma$ confidence level.

---

[1]In Ref. [18] the simplified model doesn't consider the effect of the existing uncertainties on each one of the assumptions and parameters' value, and without considering other possible alternatives.

SciPost Phys. Proc. **12**, 021 (2023)

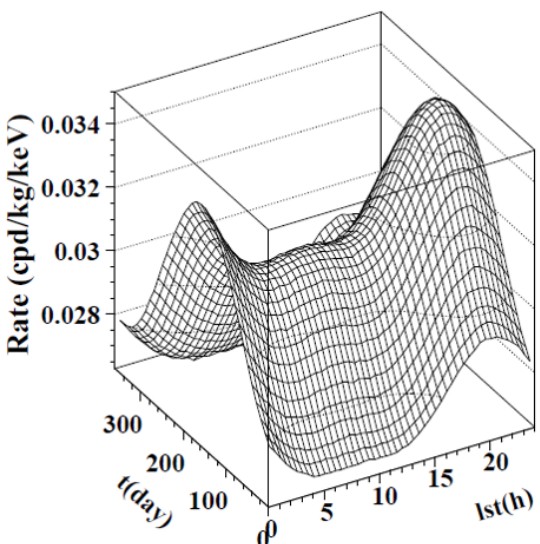

Figure 4: An expected rate as a function of sidereal time and days of the year in case of a DMp-nucleus elastic scattering with a 200 kg of a multi detector of $5 \times 5$ ZnWO$_4$ anisotropic scintillators. The model assumes a DMp mass of 10 GeV, with a cross section on nucleon of $5 \times 10^{-5}$ pb (dominated by a spin-independent coupling constant and a scaling law of the DM-nucleus elastic cross section very simple, see Ref. [18]). All the details of the framework considered are carefully illustrated in Ref. [18]. The plane identified by the day-axes and the counting-rate-axes shows the profile of the annual modulation amplitude for a fixed sidereal day. This figure is being reused from Ref. [30] with permission (license number 1273426-1)

We underline that the ZnWO$_4$ is one of the most radiopure crystal scintillators; however, we are still working to increase the radiopurity and the size of the crystal scintillator to identify the best condition to produce ZnWO$_4$ crystal scintillators and to optimize the optical properties for a large crystal detector. Moreover, we are investigating the light yield of ZnWO$_4$ with the temperature operation down to $-40$ °C. We expect that light yield should increase at a lower temperature [26]. If the effect is confirmed, we could decrease the detector's software energy threshold and contemporarily get a better energy resolution. At the same time, we plan to perform new measurements to study the Q.F. energy dependence at lower energy with respect to those reported here.

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
