# Peer review of "Dark Matter Directionality Approach Using ZnWO4 Crystal Scintillators"

_SciPost Physics Proceedings, doi:SciPost Phys. Proc. 12, 021 (2023)_

## Round 1 · Referee Report · Anonymous (Referee 2) · 2022-11-8

Weaknesses

This is a summary of already published works. The references are mostly to their previous works. The proof of principle of such approach was already proved in 2020, but no future plans are presented with a timeline in this manuscript.

Report

Requires revision and improvement.

Requested changes

- "The Q.F.’s reported in Ref. [21] are in suitable congruity with those of Ref. [20], as pointed out in Fig. 1-right." please explain the difference lead to this result.
- Is there is something new you can add? Future plans? Current status?
- Please mention other studies on this subject, explain the difference (if there is).

  • validity: -
  • significance: -
  • originality: -
  • clarity: -
  • formatting: -
  • grammar: -

Author:  Vincenzo Caracciolo  on 2022-11-24  [id 3065]

(in reply to Report 2 on 2022-11-08)
Category:
answer to question

Answers to the REVIEWER

We thank the Referee for the advice to improve the paper's quality. In the following, we attach the answers to the questions and comments.
All the revisions made to the manuscript have been marked up using the blue colour as a track change to be easily viewed by the editors and the reviewer.

---

## Round 1 · Referee Report · Anonymous · 2022-11-8

Weaknesses

Answer: This is a summary of already published works. The references are mostly to their previous works. The proof of principle of such an approach was already proved in 2020, but no future plans are presented with a timeline in this manuscript.
As the talk given at the IDM22 conference, the proceeding summarises the results achieved in several years of R&D and measurements about the topic of the Dark Matter Directionality Approach Using ZnWO4 Crystal Scintillators. In fact, besides the results achieved in 2020, we also reported the latest developments after a very long R&D in order to increase the optical and scintillation features of ZnWO4 crystals; the relative paper has been recently published (2022). In addition, following the referee's suggestion, we improved the manuscript with a new version that addresses plans. Following the referee's comments, we also mention other studies on this subject, briefly explaining the differences.
Report
Requires revision and improvement.
Requested changes

Point 1. "The Q.F.’s reported in Ref. [21] are in suitable congruity with those of Ref. [20], as pointed out in Fig. 1-right." please explain the difference lead to this result.
Answer point1:
The measurements achieved in the refs. [26] and [27] of the new version of the manuscript, as pointed out by the referee, are in good agreement. However, some deviation could be even possible because of impurities or the entire procedure of building crystals. Such an anisotropic capability is mainly because, in this type of crystal scintillator, the highly ionizing particles, as nuclear recoils, generate different amounts of excitons in such a direction with respect to the crystallographic planes.

Point 2. Is there is something new you can add? Future plans? Current status?
Answer point 2: We have added a new paragraph at the end of the manuscript and highlighted it in blue.

Point 3. Please mention other studies on this subject, explain the difference (if there is).
Answer point 3: We have added a new paragraph at the end of the manuscript and highlighted it in blue.

Attachment:

idm22_crc.pdf

---

## Round 2 · Referee Report · Anonymous · 2022-11-30

Report

Can be published.

---

## Editorial Decision

published